# Red flags Presented in Children Complaining of Headache in Paediatric Emergency Department

**DOI:** 10.3390/children10020366

**Published:** 2023-02-13

**Authors:** Rosaura Conti, Giorgia Marta, Lotte Wijers, Egidio Barbi, Federico Poropat

**Affiliations:** 1Department of Medicine and Surgery and Health Sciences, University of Trieste, 34127 Trieste, Italy; 2Faculty of Health, Medicine and Life Sciences, Maastricht University, 6211 LK Maastricht, The Netherlands; 3Institute for Maternal and Child Health, IRCCS “Burlo Garofolo”, 34137 Trieste, Italy

**Keywords:** headache, red flags, nocturnal awakening, occipital pain, vomiting

## Abstract

This study aimed to determine how common are specific red flags of life-threatening headache (LTH) among children with complaints of headache in the emergency department. A retrospective study was conducted over five years, including all patients aged < 18 years who presented for a headache to a Pediatric Emergency Department. We identified patients with life-threatening headaches and compared the recurrence of the main red flags (occipital location, vomit, nocturnal wake-up, presence of neurological signs, and family history of primary headache) to the remaining sample. Two-thousand-fifty-one children (51% female, 49% male) were included. Seven patients (0.3%) were diagnosed with a life-threatening headache. In the analysis of red flags, only the presence of abnormal neurological evaluation and vomiting was found to be more common in the LTH sample. No statistically significant difference was found for nocturnal awakening or occipital localization of pain. Urgent neuroradiological examinations were performed in 72 patients (3.5% of cases). The most common discharge diagnosis was infection-related headache (42.4%), followed by primary headaches (39.7%). This large retrospective study confirms the most recent literature suggesting that night awakenings and occipital pain are common symptoms also associated with not-LTH. Therefore, if isolated, they should not be considered red flags.

## 1. Introduction

Headache is a common symptom during childhood, affecting 12.8% of school-age children and increasing progressively to 49% in the teenage years [1]. The disease is classified as primary or secondary depending on the underlying etiology, according to the international classification [2].

In the pediatric setting, headache is generally a benign and self-limiting condition, which can be diagnosed based on history and physical examination. However, this symptom may underlie serious or even life-threatening diseases.

Choosing when and to whom to perform neuroimaging is the daily dilemma faced by emergency department physicians. The widespread use of diagnostic imaging is limited by the irradiation biological cost of CT scans and by the limited real-time availability of MRI, both in terms of imaging execution and the need for sedation in younger patients [2]. Some studies showed an increased risk of cancer in childhood and adulthood for pediatric irradiation up to triple the risk of leukemia and brain cancer [3,4,5]. Moreover, given the high prevalence of the disease and the very small number of secondary causes requiring specific treatment, such as abscesses, thrombosis and tumors, indiscriminate use of imaging would imply significant costs both for families and health systems. In fact, in the setting of an a priori very low diagnostic yield, the cost-benefit ratio of such an approach would be highly questionable. For this reason, physicians rely at first on clinical red flags to identify patients at risk of concerning conditions. The most common are headache on or soon after awakening in the morning, night awakening, vomiting, occipital pain, visual distortion, abnormal neurological signs, headache precipitated with Valsalva maneuver, worsening of school performance, behavioral changes, poor stature growth, a pattern of very severe acute or increasingly worsening pain and lack of family history of primary headache [6,7].

However, in the last decades, different studies found conflicting results on the predictive value of these alarming signs, showing both a low sensitivity and low specificity in predicting ominous secondary causes [8,9,10,11].

Among these, the role of some specific signs, such as night awakening and occipital pain, has already been questioned, being suggested by some studies but denied by others [11,12,13].

Consequently, doctors facing children with headaches are frequently stuck in a sort of Hamletic doubt.

Our study aimed to determine how common are specific red flags of life-threatening headache (LTH) among children with complaints of headache in the Pediatric Emergency Department (PED), comparing the prevalence of them between children with alarming conditions and the rest of the sample. Aiming at the widest possible perspective, we enrolled all children with a headache complaint, independently of age or other concomitant symptoms, such as fever.

The secondary outcomes were to investigate the etiology and clinical characteristics of this population, subdivided into age groups, and to analyze the management and treatment undertaken.

## 2. Materials and Methods

This is a retrospective study on patients who were admitted to the PED of Maternal and Child Health “Burlo Garofolo” from January 2016 to December 2020 with a chief complaint of headache. The Emergency Department is the only pediatric facility in an area of 260.000 inhabitants, with approximately 25.000 admissions per year of patients from zero to 18 years of age, in the context of a third-level pediatric teaching hospital with round-the-clock availability of pediatric radiologists and neurologists.

The data were obtained by consulting the electronic database (“G2 system”) that assembles information from all PED visits. We included in the cohort all patients aged 0–18 years complaining of headache at the triage admission. Moreover, all the patients must have completed the entire diagnostic process with the elaboration of a final diagnosis. We excluded from the study all patients older than 18 years old attending PED for the urgent onset of the symptoms or those children who went away before obtaining a discharge diagnosis, evenly children with known structural intracranial disorders or metabolic/neoplastic disease.

For each patient, we have analyzed every clinical recorded data and collected the following variables: age, sex, the score of pain at the triage, characteristics of headache, associated symptoms and signs, recurrence of the disease in the family, diagnostic workup, management, discharge diagnosis and eventual readmission in the PED for the same reason. Through the “G2 system”, the multimedia health archive, it was possible to access the scheduled and unscheduled follow-up visits, tests and hospital admissions performed after discharge and up to 2022. The data of all enrolled patients were revised from this perspective.

Patients were stratified by an arbitrarily chosen age criteria into three different subgroups, under 6 years (preschool age), from 7 to 11 (primary school age) and aged 12 years old and older (adolescents).

Final diagnoses were classified according to the International Classification of Headache Disorders (ICHD-3) and divided into the following subgroups: primary headache (migraine, tension-type headache), secondary headache due to concurrent infections, vascular diseases (ischemic or hemorrhagic stroke, deep sinus venous thrombosis) or attributed to nonvascular urgent disease (neoplasm), post-traumatic headache, headache as a sign of psychiatric symptoms and others (all diagnosis do not categorize in the previous classification). Focusing on psychiatric diseases, we have considered particularly Somatic Symptoms and Related Disorders, diagnosed according to the DSM-V [14], a group of diseases in which youths have physical symptoms that are either very distressing or result in significant disruption of their daily functioning, as well as excessive thoughts, feelings, and behaviors regarding those symptoms.

Life-threatening headache (LTH) was defined as any disorder needing prompt intervention in the PED, as already reported in previous studies [15]. Rare causes are generally considered, but these are diseases associated with high mortality and morbidity. The main conditions to worry about include central nervous infection, hydrocephalus, intracranial hemorrhage, venous sinus thrombosis, ischemic stroke, and brain tumors [16,17,18].

Among the different red flags described in the medical literature, we considered: occipital location, vomit, nocturnal wake-up, presence of neurological signs, and family history of primary headache, the last deemed as a possible protective factor.

Because this study used pre-existing, deidentified data, the Institutional Review Board deemed it exempt from Ethical Committee approval. According to Italian Law, the Authorization to Process Personal Data for Scientific Research Purposes (Authorization no. 9/2014) declared that retrospective archive studies that use ID codes, preventing the data from being traced back directly to the data subject, do not need ethics approval. According to the standard Research Institute practice, informed consent was required and signed by parents at each visit, in which they agree that “anonymous clinical data may be used for clinical research purposes, epidemiology, study of pathologies and training, to improve knowledge, care and prevention” [19].

Statistical analysis was performed using the website OpenEpi19. We expressed the continuous data as means and range, the categorical data as absolute frequencies and percentages, and performed comparisons using Student’s *t*-test or Fisher’s exact test for continuous and categorical variables, respectively. Statistical tests were two-sided, and a *p*-value of less than 0.05 was considered statistically significant.

## 3. Results

Two-thousand-seventy-seven children complaining of headaches were assessed at the Emergency Department (ED) in the study period between 2016 and 2020. Of these, 8 patients were excluded for age > 18 years, 4 for triage error and 14 for hospital abandonment before medical evaluation. The general characteristics of the population are shown in Table 1.

In seven children (0.3%), the headache was a secondary expression of severe disease. Causes of life-threatening headaches (LTH) were hemorrhagic stroke (one patient), ischemic stroke (one patient), sinus deep venous thrombosis (one patient), bacterial meningitis (one patient), and cerebral neoplasms (three patients).

Red flags (Table 2 and Table 3), excluding a family history of headaches, were described 649 times, 31% of the whole sample. Considering each red flag, 58 patients had abnormal neurologic findings during clinical evaluation (examination of mental state, meningeal signs, cranial nerves function, pathological cerebellar signs, balance and coordination, motor and sensory skills and abnormal osteotendinous reflexes). Specifically, in the LTH group, two cases were reported: one was a child with ataxia, and the other indicated a paresthesia of the body’s half.

Among all the other diseases, 56 patients were counted, with a result statistically significant (*p* < 0.01).

Remarkably, abnormal neurological signs were also reported in 4 patients out of 26 with a diagnosis of somatic symptoms and related disorders. Two of these patients showed an ataxic gait and incoordination, one patient complained of an alteration of tactile sensitivity, and the last had an abnormal Mingazzini sign in the upper arms. Eventually, all abnormal findings observed in these children by the emergency doctor were deemed counterfeit by the neurologist evaluation, disappearing with distraction techniques or because they were not congruent.

Family history of first-degree relatives with headaches was common in both LTH and not-LTH, without statistical differences. Interestingly, children with a diagnosis of somatic symptoms and related disorders had the highest rate (84.6%), more than twofold of primary forms.

Considering the etiology of headaches in the cohort, the most common discharge diagnosis was infection-related headache (42.4%), followed by primary headache (39.7%). Stratifying data by age groups, the symptom attributed to infection represented two-thirds of all causes in the youngest cohort, whereas the primary form increased progressively, becoming the first cause of pain in adolescents (47.1%). The prevalence of headaches related to somatic symptoms and related disorders increased linearly with age, even if without statistical significance (Figure 1).

Among primary headaches, according to age, migraine increases, while tension headache lowers (Figure 2).

The pain was recorded at the triage’s admission according to the VAS score, and the average value observed was 4.3, with no difference among various etiologies. Patients who received drug therapy at the PED were 636 (31%). Painkiller drugs used were mainly non-steroidal anti-inflammatory drugs (NSAIDs) (445 equal to 70% of total) and acetaminophen (188, 29.5%). Opioids (10, 1.5%) and triptans (3, 0.5%) were rarely prescribed. Association with anti-emetic drugs was prescribed in 167 patients (26.2%); 67 children received metoclopramide and 100 ondansetron.

During attendance in the PED, 272 patients (13.3%) were also evaluated by a child neurologist. In 226 children (11%), a fundus oculi investigation was performed to rule out papilledema, which was found positive in 11 cases.

Urgent neuroradiological examinations were performed in 72 patients (3.5% of all cases), 19 were Computed Tomography scans (CT), and 53 were Magnetic Resonance Imaging (MRI). The investigation had a positive finding in 13 of 53 MRI and 5 of 19 CT studies, with a positive rate of 24.5% and 26.3%, respectively.

Four patients (5.5%) had incidental benign abnormalities (“incidentaloma”). None of the neuroimaging studies performed eventually during hospital admission or in the follow-up in the rest of the cohort have shown any pathological findings.

On discharge from PED, 363 of 815 children (44.5%) with primary headache were referred to neuropsychiatric follow-up. Among these patients, 38 (10.4%) had a new diagnosis of somatic symptoms and related disorders. The number of repeat visits for the same symptom was 141 (6.9%).

## 4. Discussion

This retrospective study reports the second-largest clinical series of headaches in PED available in the literature. It shows a prevalence of headache of 1.8% among all causes of visits, with an incidence of life-threatening headaches (LTH) of 0.3% of children. Previous studies have shown a linear growth increase of children seeking care in PED for this kind of symptom in the last decades without a proportional increase in hospital admissions [20,21,22]. These data confirm that headache rarely underlies an urgent condition [1,9,23,24]. In other words, according to these figures, the chance for an emergency room doctor to face an urgent headache is pretty low, equal to one case in every 350 visits. However, a missed or delayed diagnosis of even a single case of a life-threatening condition carries a high burden of consequences for patients, families, and physicians. For this reason, the emotional burden of a possibly missed diagnosis should not be underestimated when considering a rationale and cost-benefit approach to the differential diagnosis of headache.

A careful history and an accurate physical examination are the mainstays of every pediatrician’s practice, but the real-life contest of an overcrowded ED with time restraints, emotional pressure and relational issues in terms of communication between colleagues at every shift change may interfere with every best practice.

In order to reduce the risk of missing important diagnoses, red flags have been identified and applied when managing a child with a headache, but there is no agreement among studies with a lack of strong data [10,25]. While this is a time-honored approach, hard evidence is still lacking for some of these signs.

We have tried to weigh up the role and the spread of some of these alarm bells among patients complaining of a headache in PED, particularly the site of pain, nocturnal awakenings, vomiting, altered neurological examination and family history of primary headache.

A relevant finding of this study is that we have found that almost one-third of the whole sample of patients reported one or more red flags, pointing out how ubiquitous these signs or symptoms are in contrast to the rarity of emergent intracranial abnormalities. As in previous studies [9,10], we reiterate the need to more clearly understand how or which red flag findings should be used to decide whether emergent neuroimaging is indicated.

We believe that the most relevant finding of this study is the absence of a correlation between LTH and nocturnal awakening, a relatively common symptom complaint also in the general population with headache (9.2%). Common wisdom suggests that a pain that awakens from sleep is indicative of increased intracranial pressure, consequent to a redistribution of the forces of gravity from an upright to a supine position. Nevertheless, two previous studies have already shown that this sign was present in almost 25% of cases of primary headache [16,26]. Moreover, in Lanphear’s study of presenting symptoms of pediatric brain tumors diagnosed in the emergency department, 13.8% of patients reported morning headaches, 12.1% occurred in the evening, and 70% were of unknown timing [27].

Likewise, pain in the occipital area, a localization previously included as an element of alarm for LTH headache [2,28,29], was not related to urgent conditions in our series. There are several studies [11,27] that associate the occipital location and urgent disease in children attending PED, but in both samples, all patients with severe disease and occipital site of pain also had an abnormal neurologic examination. A retrospective cohort study [30] of pediatric patients in an outpatient child neurology office did not find an association between occipital headaches and serious intracranial pathology, showing that 7% of headaches were solely occipital and a total of 14% of headaches also included occipital pain. In light of these data, it may be suggested that isolated occipital pain, even if less common than the frontal location, should not worry per se.

As expected, vomiting and neurological abnormalities have shown to be strong predictors of LTH in our study, as in the previous ones [9,10,11]. The possible add-on of our study is that it underlines that these are not such rare symptoms, also in the non-LTH population.

Abnormal neurological signs remain the leading finding to push emergency doctors to rule out severe diseases [25,29], but interestingly we have found inconsistencies during the neurological exam performed by the neurologist in almost one-fifth of patients with somatic symptoms and related disorders. From this perspective, we can speculate that a specialist consultation is crucial to rule out actual neurological involvement and avoid useless diagnostic procedures.

Focusing on familiarity, a history of headaches in the family is quite common in both groups, and it is not useful to discern LTH and non-LTH, as already shown by other studies [31]. Interestingly, in our study, children with somatic symptoms and related disorders had the highest family recurrence rate for headaches.

Moving away from urgent conditions, we have also studied the correlation between patients’ age and type of headache. As expected, the etiology of headaches changes according to age. If we consider the entire population, infectious forms are as common as the primary type, as shown by other previous studies [1,11,23]. Subdividing the sample by age group, secondary headache due to infection represents the large majority of the cases (two-thirds) in preschool children, while in the oldest cohort, primary headache is the more frequent cause, similar to what is usually reported in adults [32]. Tension-type was the most common form in all ages, with the frequency of migraine progressively increasing with grown-up patients. The difference among age groups could explain conflicting data from the literature, where some studies describe the primary form as most prevalent, and others find secondary headaches to be more common [1,26].

Interestingly we have also found that part of our children (10.4%), classified at first as affected by a primary form at the discharge of ED, during neuro-psychiatric follow-up received the diagnosis of a somatic symptom and related disorders. This finding highlights that in any form of chronic or recurrent pain in adolescence, psychosomatic conditions should be considered [33]. The relationship between anxiety, depression and headache is clearly established. Indeed a population-based study of 8-year-old Finnish schoolchildren found an increase in psychosomatic complaints in a decade, most notably expressed as headache and abdominal pain, and showed how strongly they were correlated with measures of depression and anxiety [34].

The rate of neuroimaging (3.5%) in our population was lower than reported in most other studies [23,28,35,36,37,38]. Actually, several previous retrospective studies have shown different rates of imaging, from 6.3% to 44% of children presenting to the ED with headaches [23,36]. This wide variation reflects, once more, the lack of consensus or formal recommendations regarding the management of headaches in the acute care setting [39]. Regardless, our rate is one of the lowest reported in the literature, and it is not easy to understand the reason. We think that an explanation can be found in the primary care system organization in Italy. In the Italian National Health System, every child is entrusted to a family pediatrician up till the age of 14 years. For this reason, the emergency doctor is probably more confident that a strict and proper follow-up is guaranteed for each patient. As a matter of fact, a previous study [40] comparing Italian and English admission rates after PED evaluation already showed a statistically significant trend toward lower figures of admission in the Italian system. This seems to suggest that the availability of a dedicated family pediatrician, compared to an adult’s general practitioner, may facilitate a prompt discharge from the ED.

The prevalent radiological investigation was MRI (two-thirds of cases), and the rate of pathological findings was 25% considering all tests performed, like the rate detected in the previous series (15–25%) [35,36,37,38]. No important diagnoses were lost in the PED, as no patient undergoing neuroimaging after discharge showed new findings relevant to the treatment. We must underline, otherwise, as a downside of the easier feasibility of MRI is the growing prevalence of “incidentalomas”, incidental discoveries not congruent with the symptom. The incidence in our sample was 7.5%, but in literature, a figure close to 25% is reported [41,42], with the subsequent heavy burden of non-urgent investigation without clinical relevance and the distressed family resulting.

Considering the use of painkillers, the most prescribed drug was ibuprofen, while a low rate of children received acetaminophen, metoclopramide, or ondansetron. In the guidelines, there is a strong recommendation regarding the efficacy and safety of non-steroidal anti-inflammatory drugs (NSAIDs). Ibuprofen particularly is comparable to paracetamol in reducing pain, but the first shows better rapidity of action with moderate levels of efficacy in resolving the episode after two hours [43]. We remark that only 1.5% of patients received opioids in this cohort, in conflict with other reports where the prescription rate is close to 30% of children [44]. Narcotics are considered a potential adjunctive therapy option for status migrainosus in adults, but there is no evidence to support their use in children or adolescents with migraine [45]. Additionally, their prescription has been associated with a greater readmission rate and longer lengths of stay [44].

The main limitations of this study are the retrospective collection of the data from a single pediatric emergency department and the inherent limitations of the database. Moreover, the small numbers of LTH found grant our study a low predictive statistical power. The points of strength are the numerosity of the survey, one of the largest in the available literature, which has allowed us to subdivide the population into age cohorts and the additional information from the follow-up of the children.

## 5. Conclusions

Red flag findings are common in otherwise healthy children presenting with headaches to the PED. This large retrospective study confirms recent literature suggesting that night awakenings and occipital pain, isolated, are symptoms also associated with not life-threatening headaches. Abnormal neurologic findings, as well as vomiting, are the strongest predictors of severe diseases needing prompt intervention, but neurologist consultation can help to rule out incongruent symptoms. Considering all children attending PED for headaches, a secondary form due to infection represents the large majority of the cases (2/3) in preschool children, while in the oldest cohort, primary headache is the more frequent cause, similar to what is usually reported in adults.

## Figures and Tables

**Figure 1 children-10-00366-f001:**
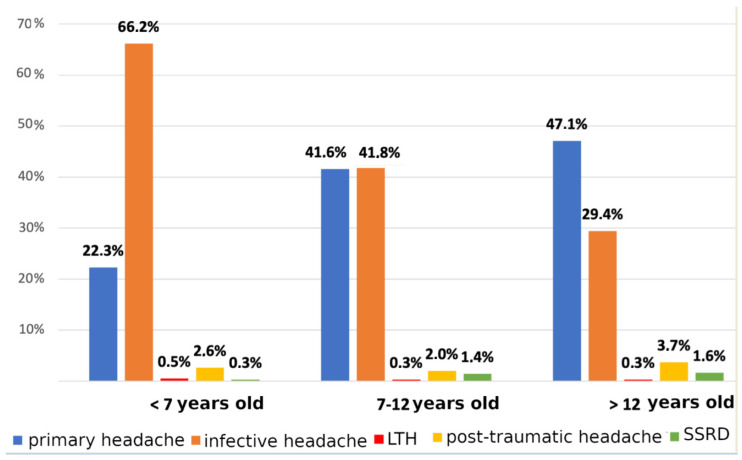
Age distribution of not life-threatening headaches. LTH: life-threatening headache; SSRD: somatic symptom and related disorder.

**Figure 2 children-10-00366-f002:**
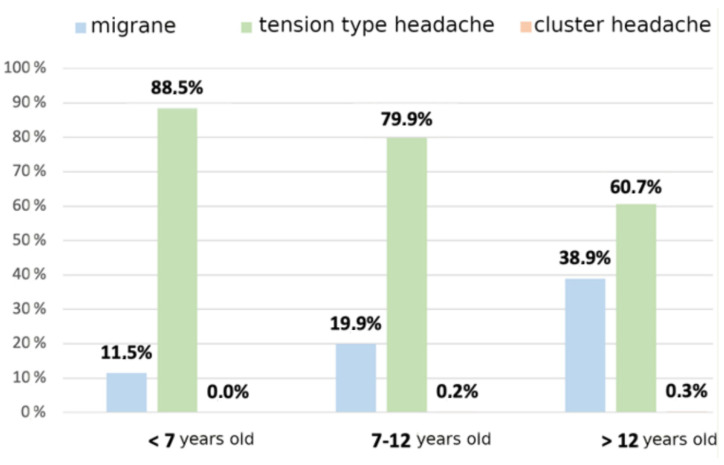
Age distribution of primary headaches.

**Table 1 children-10-00366-t001:** General characteristics of the sample.

Total	2051
	10.4 (+/− 3.9)
Age <7 years 7–12 years >12 years	391 (19.1%) 979 (47.7 %) 681 (33.2%)
Sex Male Female	1006 (49%) 1045 (51%)
Causes	
Primary headache	815 (39.7%)
Secondary headache	
Infective	
Post-traumatic	869 (42.3%)
Somatic disorder	55 (2.7%)
Others	26 (1.3%)
	286 (14%)

**Table 2 children-10-00366-t002:** Frequency of red flags in patients with or without life-threatening headaches.

Red Flags			
	LTH	Not-LTH	*p*-Value
abnormal neurologic findings	2 (28.6%)	56 (2.7%)	<0.01
vomit	6 (85%)	319 (5.6%)	<0.01
nocturnal awakening	1 (14.3%)	189 (9.2%)	*ns*
occipital site	0 (0%)	76 (3.7%)	*ns*
family history	1 (14.3%)	340 (16.6%)	*ns*

**Table 3 children-10-00366-t003:** Frequency of red flags in patients with different types of non-life-threatening headaches.

	Primary Headache	Infective Headache	Post-Traumatic Headache	Somatic Disease
abnormal neurologic findings	25 (3.1)	8 (0.01)	2 (3.6)	4 (15.4)
vomit	108 (13.3)	173 (19.9)	6 (10.9)	3 (11.5)
nocturnal awakening	113 (13.9)	61 (7)	0 (0)	4 (15.4)
occipital site	45 (5.5)	173 (19.9)	2 (3.6)	0 (0)
family history	267 (32.4)	-	-	22 (84.6)

## Data Availability

The data presented in this study are available on request from the corresponding author.

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
