# Peer review of "Red flags Presented in Children Complaining of Headache in Paediatric Emergency Department"

_children, 2023, doi:10.3390/children10020366_

Round 1

Reviewer 1 Report

Thank you for the opportunity to review this work. I appreciated the efforts made by the authors to detect how common the specific red flags of life- 9 threatening headache (LTH) among children with complaints of headache in the emergency 10 departments.

The manuscript is well written. But the exclusion & inclusion criteria should be added to the methodology part.

the introduction section should be improved to clarify the goal of the research and should take the journal formate 

Author Response

Thank you very much for your advices. We have added exclusion and inclusion criteria in the materials and methods section (red part). Moreover we have changed the introduction part to clarify the goal of our study.

Reviewer 2 Report

Line 55- minor issue, but I assume you mean that this third category is ages 12 and older (versus over 12, which would leave 12 year-olds without a group)?

Line 66-"familiarity with primary headache" is not a term I'm used to hearing. I assume you mean previous personal history of headache, versus new-onset headache? Clarifying this would be helpful.

It would be helpful to specifically list the abnormal neuro findings that the 2 patients with LTH had.

Line 100- please define "somatic disturbance", as this is not a term that is commonly used, at least in the US. Do you mean functional neurological disorder?

Line 103- "family recurrence rate"- what does this mean? I assume you mean family history of migraine, but please define (eg-family history of first degree relative with migraines)

Line 104- please clarify- I think you  mean you found in your study that patients with primary headaches had family history of migraine, but it's unclear to me if you mean this, versus that previous studies have found this (if you mean previous studies, you would need a reference).

Line 124- again, please define somatic disease.

Author Response

Thank you for your tips and notes. 
In the line 55 we meant 12 years old and older [now line 72]. In the line 66, as well as in line 103, when we wrote about  “familiarity with primary headache” or “family recurrence rate” we meant the presence of family history of primary headache.
With regard to “somatic disturbance” (line 100 or 124, now 77) we referred to “somatic symptom and related disorder” in the DSM V classification.

Best regards

federico poropat